# Ubiquitin-Specific Protease 2 (OsUBP2) Negatively Regulates Cell Death and Disease Resistance in Rice

**DOI:** 10.3390/plants11192568

**Published:** 2022-09-29

**Authors:** Ruirui Jiang, Shichen Zhou, Xiaowen Da, Tao Chen, Jiming Xu, Peng Yan, Xiaorong Mo

**Affiliations:** State Key Laboratory of Plant Physiology and Biochemistry, College of Life Science, Zhejiang University, Hangzhou 310058, China

**Keywords:** OsUBP2, rice, ROS accumulation, leaf blast resistance, bacterial blight resistance

## Abstract

Lesion mimic mutants (LMMs) are great materials for studying programmed cell death and immune mechanisms in plants. Various mechanisms are involved in the phenotypes of different LMMs, but few studies have explored the mechanisms linking deubiquitination and LMMs in rice (*Oryza sativa*). Here, we identified a rice LMM, *rust spots rice* (*rsr1*), resulting from the mutation of a single recessive gene. This LMM has spontaneous reddish-brown spots on its leaves, and displays enhanced resistance to both fungal leaf blast (caused by *Magnaporthe oryzae*) and bacterial blight (caused by *Xanthomonas oryzae* pv. *oryzae*). Map-based cloning showed that the mutated gene in *rsr1* encodes a Ubiquitin-Specific Protease 2 (OsUBP2). The mutation of *OsUBP2* was shown to result in reactive oxygen species (ROS) accumulation, chloroplast structural defects, and programmed cell death, while the overexpression of *OsUBP2* weakened rice resistance to leaf blast. OsUBP2 is therefore a negative regulator of immune processes and ROS production. OsUBP2 has deubiquitinating enzyme activity in vitro, and the enzyme active site includes a cysteine at the 234th residue. The ubiquitinated proteomics data of *rsr1* and WT provide some possible target protein candidates for OsUBP2.

## 1. Introduction

Lesion mimic mutants (LMMs) exhibit constitutive cell death in the absence of exogenous pathogens, similar to the HR when pathogens invade plants, and most LMMs showed increased resistance to pathogens compared to WT [1,2]. LMMs have been reported in a variety of plants, such as *Arabidopsis thaliana* [3], maize (*Zea mays*) [4], and rice [5,6,7]. Studying the genes responsible for the LMMs is important for the elucidation of the molecular mechanisms of disease resistance in plants. Researchers have cloned multiple LMM genes in rice using forward and reverse genetic methods. *Spotted Leaf 35* (*SPL35*) encodes a protein containing a CUE (coupling of ubiquitin conjugation to endoplasmic reticulum [ER] degradation) structural domain. The *spl35* mutant exhibits reduced chlorophyll, increased H_2_O_2_, upregulated expression of defense-related marker genes, and enhanced resistance to rice fungal and bacterial pathogens [3]. Both *SPL11-interacting Protein 6* (*Spin6*) RNA interference (RNAi) lines and T-DNA insertion mutants resulted in PCD, ROS accumulation, and the increased expression of genes related to the defense response [8]. *Early Lesion Leaf 1* (*ELL1*) encodes a cytochrome P450 monooxygenase, which localizes to the ER. In the *ell* mutants, the chlorophyll content was decreased, the chloroplasts were degraded, and the photosynthetic protein content was decreased, suggesting that ELL may be involved in chloroplast development [9].

Plants have evolved a two-layered immune system to protect themselves from pathogens. The first layer of the immune system is known as the pathogen-associated molecular pattern (PAMP)-triggered immune (PTI) response, in which pattern recognition receptors (PRRs) located on the cell membrane recognize PAMPs and activate a series of immune defense responses to defend against pathogen invasion [10]. During the co-evolution of host–microbe interactions, pathogens have evolved a class of effectors that can be secreted into plant cells. Those effectors inhibit the plant immune response by attacking the PTI signaling pathway, enabling the survival of the pathogen in the plant. Meanwhile, plants have evolved a class of resistance (R) proteins that directly or indirectly monitor pathogen effectors and activate a stronger immune response, effector-triggered immunity (ETI) [11]. ETI is an accelerated and amplified PTI response, usually associated with cell death at the site of infection, known as the hypersensitive response (HR) [12].

Reactive oxygen species (ROS) are highly reactive reduced forms of oxygen molecules, including superoxide anion (O_2_^–^), hydrogen peroxide (H_2_O_2_), hydroxyl radical (·OH), and singlet oxygen (^1^O_2_). Plant ROS production largely takes place in the chloroplasts, mitochondria, and peroxisomes, and is dependent on several classes of enzymes, including nicotinamide adenine dinucleotide phosphate (NADPH) oxidase, peroxidase, and oxidase [13,14,15]. While ROS plays important roles in various processes, high levels of ROS can be highly damaging to cells; therefore, plants use enzymes such as superoxide dismutase (SOD), catalase (CAT), ascorbate peroxidase (APX), and glutathione peroxidase (GPX) to scavenge excess ROS and maintain homeostasis [16]. 

ROS plays an important role in both PTI and ETI [10]. In rice, PRRs activate various immune responses by sensing PAMPs, including the rapid and intense production of ROS by OsRbohB and other ROS-producing components, to trigger PTI in rice. In the second layer of defense, plant intracellular immune receptors directly or indirectly recognize specific pathogen effectors to induce ETI. In this case, rice activates a series of signaling pathways leading to the HR, which involves a ROS burst, callus deposition, the induced expression of pathogen-related genes, and programmed cell death (PCD) [17,18,19]. 

The posttranslational modifications of proteins play key regulatory roles in most cellular signaling events. Ubiquitination is a typical posttranslational modification found in eukaryotes [20]. The ubiquitin proteasome pathway has been reported to be involved in the formation of LMMs in rice. A multi-unit RING E3 ubiquitin ligase (CRL), consisting of Cullin3 (CUL3), RING-box protein 1 (RBX1), and BTB proteins, was shown to be involved in the plant immune response [21]. The *cul3a* mutants develop reddish-brown lesions on leaves at 60 days post-sowing (dps) under greenhouse conditions or at 45 dps under field conditions. *Spotted*
*Leaf-11 SPL11* encodes a ubiquitin ligase, and the *spl11* mutant displays iron rust spots in the late tillering stage through to maturity. The *spl11* mutant was shown to have lost the ability to regulate NADPH oxidase activity, resulting in the accumulation of ROS [22,23]. 

Deubiquitination is the opposite of ubiquitination. Deubiquitinating enzymes (DUBs) dissociate ubiquitin from ubiquitin–protein conjugates [24], and some plant DUBs have been shown to be involved in immune response processes. The UBIQUITIN-SPECIFIC PROTEASEs (UBPs) AtUBP12 and AtUBP13 are functionally redundant, and are both able to hydrolyze the K48-linked ubiquitin chain. The RNAi mutants of *AtUBP12* and *AtUBP13* displayed an enhanced resistance to the pathogen *Pseudomonas syringae* pv. *tomato* (*Pst*) DC3000 [25]. The deubiquitinase PICI1 (PigmR-interacting and chitin-induced proteins) is an immune hub of PTI and ETI in rice. PICI1 deubiquitinates and stabilizes methionine synthase, activating methionine-mediated immunity mainly through the biosynthesis of methionine. PICI1 is targeted for degradation by blast fungal effectors, including *AvrPi9*, to inhibit PTI [26]. A deubiquitinating enzyme OsLMP1 (Lesion Mimic Phenotype 1) regulates immune responses in rice through the epigenetic modification of H_2_B histones [27]. The mutation of *OsLMP1* resulted in lesions on the leaves and an enhanced resistance to rice bacterial blight (caused by *Xanthomonas oryzae* pv. *oryzae* [*Xoo*] PXO99). These mutants had elevated levels of histone H_2_B ubiquitination and histone H3-K4me2/3 methylation, which directly activated the salicylic acid (SA) biosynthesis genes *OsPAL6* (*Phenylalanine Ammonia-lyase Genet 6*) and *OsPAL7* (*Phenylalanine Ammonia-lyase Genet 7*) to improve plant disease resistance [27].

In the present study, we identified a rice LMM, *rust spots rice* (*rsr1*), the leaves of which develop reddish-brown rust spots. The underlying mutant gene encodes a functional DUB, OsUBP2/OsLMP1. *OsUBP2* was located at the same locus as *OsLMP1*. The mutation of *OsUBP2* led to ROS accumulation, chloroplast structural defects, and PCD, with the *Osubp2* mutant displaying an enhanced resistance to both leaf blast (caused by *Magnaporthe oryzae*) and bacterial blight (caused by *Xanthomonas oryzae* pv. *oryzae*). Overexpressing *OsUBP2* weakened the resistance to rice blast. This study established that OsUBP2 functions as a negative regulator of immunity and ROS accumulation in rice.

## 2. Results

### 2.1. Phenotypic Identification of the rsr1 Mutant

The rice *rsr1* mutant was identified in a population of M2 mutant lines of the rice cultivar *Nipponbare* mutagenized by ethyl methanesulfonate (EMS) treatment. Compared with the wild type (WT), the *rsr1* mutant developed reddish-brown lesions on the leaves at the four-leaf stage. The lesions spread from the tip of the leaf to the base, starting with the older leaves. At the seedling stage, lesions could only be observed on the leaf tips (Figure 1A,B), but were distributed over the entire leaf at the heading stage (Figure 1C,D). To verify whether the formation of lesions in the *rsr1* mutant is light-dependent, the middle sections of the leaves were covered with aluminum foil for seven days. No lesions formed in the shaded parts (Figure 1E), indicating that lesion formation in *rsr1* is light-dependent.

To determine the disease resistance of the *rsr1* mutant, both *rsr1* and the WT were inoculated with *M. oryzae* (*Magnaporthe oryzae*) isolate RB22 and the bacterial blight *Xoo* (*Xanthomonas oryzae* pv. *oryzae*) isolate P6 during the nutritional growth stage. *rsr1* showed strong resistance to both *M. oryzae* and *Xoo* (Figure 1F–I). The expression of genes related to disease resistance, including *OsPBZ1*, *OsAOS2*, *OsPR1a*, and *OsPR1b*, were significantly upregulated in *rsr1* compared with the WT, as determined using qRT-PCR (Figure 1J).

### 2.2. RSR1 Encodes OsUBP2, a Predicted Deubiquitinating Enzyme

To clone the *RSR1* gene, we crossed *rsr1* with *Kasalath*, and the F_1_ generation was selfed to obtain the F_2_ generation population. All of the F_1_ plants showed a WT phenotype with no lesion spots. By contrast, the F_2_ generation population showed segregation of traits, with a WT: mutant phenotype ratio of approximately 3:1 (Appendix A), indicating that the *rsr1* mutant phenotype is controlled by a single recessive gene. We designed 240 pairs of SSR primers based on the SSR polymorphisms between *Nipponbare* and *Kasalath*. Initially, using 25 homozygous recessive F_2_ individuals, *OsRSR1* was mapped onto chromosome 9 between the SSR markers RM107 and RM6174 (Figure 2A,B). We then developed a series of markers (17.7 MB, 19.4 MB, and 19.7 MB; for details, see Appendix A) between the SSR markers RM107 and RM6174 and carried out further mapping analyses. In 25 F_2_ mutant progenies, we observed 6, 0, and 1 recombination events between the *rsr1* locus and 17.7 MB, 19.4 MB, and 19.7 MB, respectively (Figure 3A). Using an F_2_ population derived from a cross between *Nipponbare* and *rsr1* for MUTMAP sequencing, a base deletion (C; 1157 bp from the start codon of the coding sequence [CDS]) was identified in the first exon of gene *Os09g0505100* (Figure 2C,D). This mutation causes an open reading frame (ORF) shift in *Os09g0505100* and results in the truncation of the protein (Appendix A). Evolutionary tree and sequence alignment analyses revealed that *Os09g0505100* is closely related to AtUBP2 in *Arabidopsis* (Appendix A); therefore, in this article, the gene is named *OsUBP2* and the *rsr1* mutant is renamed *Osubp2*.

To confirm whether the *rsr1* phenotype was caused by the mutation of *OsUBP2*, we constructed a complementation plasmid, which included the coding frame of *OsUBP2* with the 3000-bp upstream sequence, and transformed it into the *Osubp2* mutant. The transgenic plants were named pOsUBP2, and their phenotype was consistent with that of the WT (Figure 2E). No lesion spots appeared on the leaves (Figure 2F), and the disease resistance of pOsUBP2 to *Xoo* did not differ from that of the WT (Figure 2G,H). We also knocked down the *OsUBP2* gene in the *Nipponbare* background using CRISPR/Cas9-mediated gene editing technology and obtained an *OsUBP2-KO* mutant with a large deletion in *OsUBP2*. The *OsUBP2-KO* mutant was phenotypically identical to the *Osubp2* mutant (Appendix A).

### 2.3. OsUBP2 Is a Functional Deubiquitination Enzyme

OsUBP2 contains two domains, a ubiquitin carboxyl-terminal hydrolase (UCH) and a zinc-finger in the ubiquitin-hydrolases (zf-UBP) domain. UCH possesses two conserved catalytic motifs, the Cys- and His-boxes (Appendix A). Within these boxes are positionally conserved Cys and His residues within the active site [28]. To confirm whether OsUBP2 has deubiquitinating enzyme activity, GST-OsUBP2, GST-OsUBP2^C234S^ (carrying a mutation of cysteine to serine at position 234), or GST were co-transformed with His-UBQ1 (ubiquitin extension protein) and His-UBQ10 (hexameric polyubiquitin) into *Escherichia coli* BL21 (DE3). Immunoblot analysis indicated that only GST-OsUBP2 could cleave ubiquitin molecules (Figure 3A,B). To investigate whether the phenotype of *Osubp2* results from the absence of its kinase activity, pOsUBP2^C234S^ (p represents the promoter of *OsUBP2*) transgenic plants were generated in the *Osubp2* background. Unlike pOsUBP2, pOsUBP2^C234S^ could not recover the lesion phenotype of *Osubp2* (Figure 3C). The *Osubp2* protein contained a shift at amino acid 844 and prematurely terminated, resulting in the complete deletion of the His-box (Appendix A). These findings show that the His-box and the cysteine residue at position 234 in the Cys-box are essential for OsUBP2 to function normally; and that OsUBP2 is a functional deubiquitinating enzyme.

### 2.4. Mutation of OsUBP2 Causes ROS Accumulation

To detect the accumulation of ROS in the *Osubp2* mutant, various histochemical staining methods were used. 3,3′-diaminobenzidine (DAB) staining showed an accumulation of H_2_O_2_ in the *Osubp2* mutant; nitro blue tetrazolium (NBT) staining showed the accumulation of O_2_**^–^** in the *Osubp2* mutant (Figure 4A). In addition, the leaves of the WT and *Osubp2* plants were immersed in solutions containing 2′,7′-dichlorodihydrofluorescein diacetate (H_2_DCFDA) to observe ROS accumulation and redox homeostasis. The oxidation state fluorescence signal of the probe was observed in the *Osubp2* mutant, but not in the WT (Figure 4C). To further explore the excessive ROS accumulation in *Osubp2* leaves, we measured the concentrations of H_2_O_2_ and the lipid oxidation product malondialdehyde (MDA), revealing both were significantly higher in *Osubp2* than in the WT (Figure 4B). 

In order to explore whether the accumulation of ROS in *Osubp2* is caused by a defective ROS-scavenging mechanism, the enzymatic activities related to ROS scavenging were analyzed in the *Osubp2* mutant and the WT. The activities of peroxidase (POD), superoxide dismutase (SOD), ascorbate peroxidase (APX), and catalase (CAT) were all increased in *Osubp2* (Figure 4E), suggesting that the accumulation of ROS in this mutant is not due to defects in the ROS-scavenging mechanism.

### 2.5. Mutation of OsUBP2 Led to Abnormal Chloroplasts and Cell Death

To characterize *Osubp2*-mediated apoptosis, a TUNEL assay was used to detect cell death in *Osubp2* and WT leaves. No positive TUNEL signal was detected in the WT; however, a strong green TUNEL signal was detected in the nuclei of *Osubp2* cells (Figure 5A), indicating intense DNA damage and likely cell death. Transmission electron microscopy (TEM) was used to observe the chloroplast ultrastructure of *Osubp2* and WT leaves. Compared with the WT, the chloroplasts in *Osubp2* contained far fewer starch granules and some irregular protrusions (indicated by a red arrow in Figure 5B), which might be degraded thylakoid structure. Both the TUNEL assay and TEM sections were performed on leaves taken from WT and *Osubp2* at 30 dps, when the lesions first appeared on the leaves of *Osubp2*. The section of the *Osubp2* leaf was taken from a region with no lesion spots. Additionally, the chlorophyll contents of the *Osubp2* and WT leaves were measured. The chlorophyll a, chlorophyll b, and total chlorophyll contents of *Osubp2* were significantly lower than those of the WT, whether in regions with or without lesion spots, but there was no difference in the carotene contents of the two genotypes (Figure 5C). Overall, the mutation of *OsUBP2* caused chloroplast morphological defects and cell death.

### 2.6. Overexpression of OsUBP2 Weakens Resistance to Rice Blast

In order to further reveal the function of OsUBP2 against bacterial blight, *OsUBP2* was overexpressed using a ubiquitin promoter. Two transgenic lines overexpressing *OsUBP2* were obtained, named OVER-9 and OVER-11. A qRT-PCR analysis revealed a 25-fold and 30-fold increase in the expression of *OsUBP2* in OVER-11 and OVER-9, respectively, compared with the WT (Figure 6D). Interestingly, the expression of the pathogen-related gene *OsPR1b* was significantly lower in OVER-9 and OVER-11 when compared with WT (Figure 6D). The leaf morphology of the overexpression lines was identical to the WT, and no lesion spots developed (Figure 6A). The overexpression lines and the WT were then inoculated with *M. oryzae* at the heading stage to observe their disease resistance 7 days post inoculation (dpi). The lesion lengths of OVER-9 and OVER-11 were longer than those of the WT (Figure 6B,C), indicating that the overexpression of *OsUBP2* decreased resistance to rice blast.

### 2.7. Expression Pattern of OsUBP2

To identify the expression pattern of *OsUBP2*, we used qRT-PCRs to test its relative expression levels in different tissues and organs in the WT, revealing that it was more highly expressed in the leaf and leaf sheath than the other tissues (Figure 7A). To further investigate the expression pattern of *OsUBP2*, transgenic plants harboring a β-glucuronidase (GUS) reporter gene driven by the *OsUBP2* promoter (the 3-kb sequence upstream of the start codon) were produced. GUS staining showed that *OsUBP2* was expressed in the leaf, root, panicle, and ligule bases (Figure 7B). To determine the subcellular localization of OsUBP2, plasmids of UBP2-fused GFP or GFP alone (as a control) were co-expressed in rice protoplasts with mCherry driven by the *35S* promoter. *OsUBP2* was found to be localized in the cytoplasm and nucleus (Figure 7C). We also transiently expressed *p35S::OsUBP2-GFP* in *Nicotiana benthamiana*, the OsUBP2-GFP fusion protein was localized in both cytoplasm and nucleus, as indicated by the overlap between GFP signals and red signals from the nucleus marker (H_2_B-mcherry) (Appendix A).

### 2.8. Ubiquitinated Proteomics Analysis of the Osubp2 and WT Plants

To find target protein candidates, label-free quantitative proteomics was used to compare the protein ubiquitination levels between *Osubp2* and the WT. A total of 394 reliable ubiquitination sites (localization probability ≥ 0.75) were identified in *Osubp2*, while 720 reliable ubiquitination sites were identified in the WT. Proteins with expression differences greater than 1.5-fold between the two genotypes and *p*-value < 0.05 were considered to be significantly differentially ubiquitinated loci. The clustering of the proteins with significantly different ubiquitin levels for *Osubp2* and WT is shown in Figure 8A. Thirty ubiquitination modification sites were significantly elevated and sixteen ubiquitination modification sites were significantly decreased in the *Osubp2* mutant compared with WT. The 26 proteins with elevated levels of ubiquitination sites in *Osubp2* could be considered candidate target proteins for OsUBP2.

## 3. Discussion

There are four main mechanisms by which LMMs are generated. First, the pathogen-associated genes may be mutated or overexpressed, disrupting the immune signaling pathways and initiating the HR [29]. This unsuppressed HR could lead to massive cell death and lesion mimic spots. The *ssi4* gain-of-function mutant in *Arabidopsis* contains an amino acid substitution in the NBS structural domain of the TIR-NBS-LRR protein SSI4, This mutation constitutively activates the immune system, resulting in the significant upregulation of the disease-related genes and subsequent PCD following SA accumulation [30]. Similar to *SSI4*, *NLS1* in rice encodes a typical CC-NB-LRR protein. The mutation of an amino acid in a non-conserved region near the GLPL motif of the NB structural domain leads to the constitutive activation of defense responses in the *nls1* mutant, including the excessive accumulation of H_2_O_2_ and SA, and the upregulation of disease-related gene expression. The *nls1* mutant develops necrotic spots on its leaf sheaths and displays an enhanced resistance to bacterial pathogens [31]. 

Another mechanism underlying the LMM phenotype is uncontrolled PCD. *OsLSD1* in rice plays a negative regulatory role in regulating PCD, while *OsLSD1*-antisense transgenic plants exhibit a LMM phenotype, with upregulated *PR-1* expression and an accelerated allergic response when inoculated with a non-virulent rice blast fungus [32]. A third mechanism underlying LMMs is metabolic pathway disorder, which can lead to the accumulation of intermediate metabolites or the blocked biosynthesis of some key enzymes in the metabolic process, thus causing the emergence of the LMM phenotype. Examples include the formation of the maize *les22* mutant [33], the *Arabidopsis rug1* mutant [34], and *rlin1* in rice [7]. Fourth, temperature and light can affect the occurrence of the LMM phenotype. High-temperature stress inhibits normal cellular processes and leads to abnormal plant growth and development. In the LMMs *spl7* [35] and *les1* [36] in rice, the lesion spots were enhanced at higher temperatures. The formation of spots was reported to be light-dependent in several LMMs in rice, with less damage found on leaves covered with aluminum foil, and fewer or no spots formed on covered areas of leaves compared with the uncovered areas [35,37]. 

### 3.1. Mutation of OsUBP2 Promotes ROS Accumulation and Cell Death

Plants need light for photosynthesis; however, too much light energy can cause significant cell damage. These symptoms were observed on several LMMs, such as *lsd1* [38], *mips1* [39], and *cat2* [40]. The phenotypes of these LMMs are light-dependent, suggesting that chloroplast activity acts as a signal that may trigger cell death through the accumulation of ROS. Lesion formation in *Osubp2* was also light-dependent (Figure 1E), and abnormal chloroplast morphology (Figure 5B) with decreased chlorophyll content (Figure 5C) was also observed. Histochemical staining experiments and H_2_O_2_ content detection demonstrated that *Osubp2* accumulated excessive levels of ROS in its leaves (Figure 4A–C). APX, SOD, POD, and CAT play important roles in the ROS-scavenging process [13], and the activities of all four were increased in *Osubp2* (Figure 4D), suggesting that the accumulation of ROS in this mutant was not the result of a defect in the ROS-removal mechanism. ROS also plays complex roles as secondary messengers in the signaling pathway of gene-regulated cellular suicide, PCD [41]. A positive TUNEL signal was detected in the *Osubp2* mutant (Figure 5A), providing further verification that the accumulation of ROS in *Osubp2* leaves caused cell death. These results suggest that mutations in *OsUBP2* lead to defective chloroplast function, the accumulation of ROS, and cell death. OsUBP2 plays a key role in ROS production and PCD, deepening our understanding of the relationship between these two processes in plants.

### 3.2. Loss of OsUBP2 Deubiquitinating Enzyme Activity Leads to the Osubp2 Phenotype

The *Osubp2* mutant forms reddish-brown rust spots on the leaves in a light-dependent manner (Figure 1A–E). The expression of *pOsUBP2* could fully rescue the *Osubp2* mutant phenotype (Figure 2E–H). OsUBP2 is a functional deubiquitinating enzyme that can hydrolyze ubiquitinated precursors AtUBQ1 and AtUBQ10 in vitro; however, this function was lost when the cysteine at position 234 in Cys-box was mutated to serine (Figure 3A,B). pOsUBP2^C234S^ could not recover the phenotype of *Osubp2* (Figure 3C). The *Oslmp1* mutant was previously reported to have truncated OsUBP2 at amino acid 868, also resulting in the deletion of the His-box and a lack of deubiquitinating enzyme activity in vitro [27]. These findings show that the His-box and the cysteine residue at position 234 in the Cys-box are essential for OsUBP2 to function normally. Deubiquitination is the antagonistic process of ubiquitination, in which DUBs dissociate ubiquitin from ubiquitin–protein conjugations [20,42]. Here, we showed that the levels of ubiquitin-linked proteins, especially K48-linked proteins, were higher in *Osubp2* than in the WT, which was even more aggravated after *Xoo* inoculation (Figure 3D). These results suggest that OsUBP2-mediated protein deubiquitination is involved in the rice immune response to *Xoo,* as well as normal rice growth and development more generally. 

Our results provide important evidence about the function of OsUBP2 in rice immunity. During the course of this study, Sun et al. 2022 independently characterized *Osubp2/Oslmp1* mutant alleles. OsUBP2/OsLMP1 was shown to epigenetically modify the SA biosynthesis pathway genes by deubiquitinating H_2_B, thereby regulating the immune response in rice [27]. They attempted to verify the interaction of OsUBP2/OsLMP1 with the H_2_B variant, but the results did not verify their interaction; therefore, they proposed that OsLMP1 and H_2_B might be connected through a new regulatory link [27]. Notably, H_2_B is a universal component of a nucleosome, and, thus, the specificity of changes in H_2_B ubiquitination only at SA biosynthetic pathway genes in the mutants further suggests that H_2_B might not be a direct target of OsUBP2. Moreover, in our study, OsUBP2 was not mainly localized in the nucleus, as previously reported; rather, our results showed a similar amount of OsUBP2 also existed in the cytoplasm (Figure 7C and Appendix A), suggesting that OsUBP2 may also function in the deubiquitination of cytoplasmic proteins. It will be of great interest to determine the precise biochemical nature and direct targets of OsUBP2 in the immune pathway, which thus far remains uncertain. Our ubiquitinated proteomics data supply possible target protein candidates, providing some clues for the further elucidation of the function of OsUBP2 and immunity mechanisms in rice. 

## 4. Materials and Methods

### 4.1. Plant Materials and Growth Condition

The EMS-mutated mutant library was generated using the *Japonica* rice *Nipponbare*. The *Indica* rice *Kasalath* was used to construct the genetic population for map-based cloning. Phenotypic characterization of the wild-type and mutant plants was performed in a growth chamber at 30 °C/22 °C (day/night) and 60% to 70% humidity, with bulb type light at a photon density of 300 μmolm^–2^ s^–1^ and a photoperiod of 14 h. The nutrient solution for hydroponics was Kimura nutrient solution described previously [43].

### 4.2. Map-Based Cloning 

For map-based cloning, the *Osubp2* mutant crossed with *Kasalath* to produce the F_1_ generation, and the F_1_ generation was self-pollinated to obtain the F_2_ generation population. Twenty-five plants with the lesion phenotype were selected from the F_2_ population for primary localization. *OsUBP2* was initially identified on chromosome 9 using SSR markers. The *Osubp2* mutant crossed with *Nipponbare* to generate F_2_ populations for MUTMAP sequencing. DNA was extracted from F_2_ populations, and genome resequencing was performed with a mean coverage of 30 times using an Illumina HiSeq2500 sequencer.

### 4.3. Construction of Vectors and Generation of Transgenic Plants

To verify that the mutation of *OsUBP2* was responsible for lesion formation in the *Osubp2* mutant, the *pOsUBP2* complementation construct was generated with the full-length *OsUBP2* sequence and the 3000-bp sequence upstream of *OsUBP2*. *pOsUBP2* was ligated into binary vector pBI101.3. Compared to the *pOsUBP2* vector, *pOsUBP2^C234S^* has a mutation that induces a change at the 234 amino acid residue from cysteine to serine in the resulting protein. The point mutation in *OsUBP2* was introduced using overlap extension PCR. To generate *OsUBP2-KO* mutants, two specific targets in *OsUBP2* were selected to construct the CRISPR-Cas9 vector. The sequences were ligated into the pYLCRISPR/gRNA vector, followed by a ligation into the pYLCRISPR/Cas9-MH vector as described previously [44].

To construct an overexpression vector for *OsUBP2*, the CDS of *OsUBP2* was amplified from *Nipponbare* cDNA and ligated into the PTCK303-Flag vector driven by the *UBIQUITIN* promoter and the pCAMBIA1300-sGFP vector driven by the 35S promoter.

To construct the *pOsUBP2::OsUBP-GUS* vector, the 3000-bp promoter sequence upstream of *OsUBP2* was amplified from *Nipponbare* genomic DNA and the CDS sequence of *OsUBP2* was amplified from *Nipponbare* cDNA. Overlap extension PCR was used to link the promoter sequence and the CDS sequence of *OsUBP2* and the product was ligated into the pBI101.3-GUS (β-glucuronidase) vector.

To perform deubiquitination experiments in vitro, GST-OsUBP2, GST-OsUBP2^C234S^, His-UBQ1, and His-UBQ10 vectors were constructed. The CDS of *OsUBP2* was amplified from *Nipponbare* cDNA and ligated into the pGEX-4T vector. The CDSs of *UBQ1* and *UBQ10* were amplified from *Arabidopsis thaliana* (ecotype Columbia) cDNA and ligated into the PET28a vector.

All constructs were transformed into mature seed-induced callus via *Agrobacterium tumefaciens* strain EHA105–mediated transformation, as described previously [45].

### 4.4. Measurement of Chlorophyll Content

Methods for the quantification of plant chlorophyll *a*, chlorophyll *b*, total chlorophyll and carotenoids have been described previously [46]. Briefly, leaf samples were cut into pieces, ground, soaked in 80% acetone and incubated in the dark until the leaves were completely decolored. The optical density of the sample solution was measured at 645, 470 and 663 nm using a spectrophotometer. Three biological replicates were measured for each sample.

### 4.5. Histochemical Staining 

To quantify hydrogen peroxide and superoxide anions, the inverted second leaves of 30-day-old rice seedlings were immersed in 0.1% DAB solution (pH = 3.8) and 0.1% NBT solution (pH = 7.8) in the dark for 48 h, respectively. The leaves were then transferred to 95% ethanol until they completely were decolored. H_2_O_2_ was also detected using a 10 μm 2′,7′-dichlorofluorescin diacetate (H2DCFDA) probe. The leaves were washed twice with 1×PBS after ten minutes of staining and analyzed using 485 ± 10 nm/535 ± 10 nm excitation/emission as described before [47]. 

### 4.6. In Vitro Deubiquitination Assay

GST-OsUBP2/OsUBP2 and His-UBQ1/His-UBQ10 were co-transformed into *E. coli* strain BL21. The *E. coli* cells were grown to OD_600_ = 0.8, cooled on ice for 30 min, and induced with IPTG at a final concentration of 0.5 mM. The cultures were further incubated at 37 °C for 3 h, and lysates were analyzed by SDS-PAGE with anti-GST and anti-ubiquitin antibodies.

### 4.7. Inoculation of Pathogens

Rice was inoculated with bacterial blight using the leaf-cutting method. The concentration of *Xoo* was adjusted to OD_600_ = 0.6 with sterile water, and the bacterial solution was sprayed on the rice seedlings with the leaf tips removed. The inoculation method for rice blast was described previously [3]. In brief, *M. oryzae* spores were collected in sterile water containing 0.05% Tween-20, the concentration was adjusted to 1 × 10^5^ spores per mL, and 10 μL was added to holes created by pricking the surface of rice leaves with a needle. Disease spot length was measured 7 days post inoculation (dpi).

### 4.8. Subcellular Localization

OsUBP2-GFP or GFP (as the control) driven by 35S promoter were co-transfected into rice protoplasts with 35s-mCherry. We also transiently expressed p35S::OsUBP2-GFP in Nicotiana benthamiana, constructs p35S::OsUBP2-GFP, p35S::GFP and p35S::H2B-mcherry were transformed into Agrobacterium tumefaciens strain EHA105 cells. Suspension cultures carrying p35S::OsUBP2-GFP or p35S::GFP and p35S::H2B-mcherry were co-infiltrated into 5-week-old N. benthamiana leaves. Fluorescence signals were detected using a LSM710 confocal laser scanning microscope (Zeiss), as described previously [48]. 

### 4.9. TUNEL Assay

Wild-type and *Osubp2* mutant leaves were embedded in paraffin and cut into thin slices. The slices were decolorized in xylene and then labeled using the TUNEL kit (C1088) from Beyotime.

### 4.10. Gene Expression Analysis

The total RNA of rice leaves was extracted with the TaKaRa MiniBEST Plant RNA Extraction Kit. cDNA was synthesized with the PrimeScript™ RT reagent Kit with gDNA Eraser using 2 μg RNA in a volume of 20 μL. qRT-PCRs were performed using TB Green^®^ Premix Ex Taq™ II (Tli RNaseH Plus) (TaKaRa) on a LightCycler 480 Real-Time PCR system (Roche), according to the manufacturer’s instructions. Relative expression levels were normalized to the reference gene *OsACTIN1*.

### 4.11. Proteomics Analysis

Total protein of the above-ground fraction was extracted from wild-type and *Osubp2* mutants at 21 days post-sowing (dps), and the protein was sent to SHANGHAI BIOPROTILE for label-free ubiquitinated proteomics analysis. The mass spectrometry database search software was MaxQuant 1.6.0.16, and the protein database was UniProt-Oryza sativa subsp. japonica (Rice)-148902-20200615.fasta.

### 4.12. Measurement of Various Antioxidant Indexes

The kits for the determination of H_2_O_2_ and MDA content, and kits for measuring POD, CAT, APX, and SOD enzyme activities were purchased from the Bioengineering Research Institute of Nanjing Jiancheng. All experiments were conducted according to the instructions, and measured 30 days after rice sowing.

## Figures and Tables

**Figure 1 plants-11-02568-f001:**
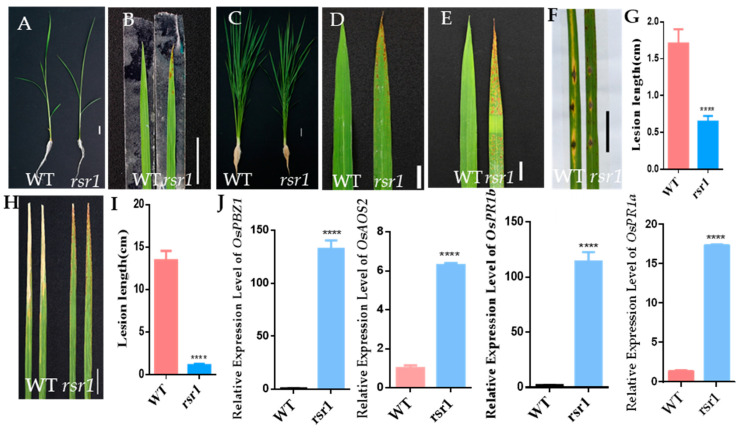
Phenotypic characterization of the *rsr1* mutant. (**A**) Wild-type (WT) and *rsr1* mutant plants at the seedling stage. Bar = 2 cm. (**B**) Leaves of WT and *rsr1* mutant plants at the seedling stage. Bar = 2 cm. (**C**) Wild-type (WT) and *rsr1* mutant plants at the heading stage. Bar = 2 cm. (**D**) Leaves of WT and *rsr1* mutant plants at the heading stage. (**E**) The phenotype of *rsr1* mutant leaf blades after shading with foil for 7 days. (**F**) Punch inoculation of *rsr1* mutant and WT plants with the compatible *M. Oryzae* isolate RB22. Leaves were photographed at 7 dpi. Scale Bar = 2 cm. (**G**) Lesion area of the inoculated leaves of *rsr1* mutant and WT plants at 7 dpi. Error bars represent + SD (*n*= 12). (**H**) Leaves of *rsr1* mutant and WT plants were inoculated with compatible *Xoo* isolate P6. Leaves were photographed at 7 dpi. Scale Bar = 2 cm. (**I**) Lesion length on the leaves of *rsr1* mutant and WT plants at 7 dpi. Error bars represent + SD (*n* = 17). (**J**) Relative expression levels of defense response genes between *rsr1* mutant and WT plants at 30 days post-sowing (dps) using the qRT-PCR assay. Data were normalized to the expression level of the reference gene *OsACTIN*. Error bars represent + SD (*n* = 3 independent pools of leaves, three plants per pool). Asterisks indicate a significant difference between the WT and *rsr1* mutant by Student’s *t*-test: **** *p* < 0.0001; qRT-PCR, quantitative real-time polymerase chain reaction.

**Figure 2 plants-11-02568-f002:**
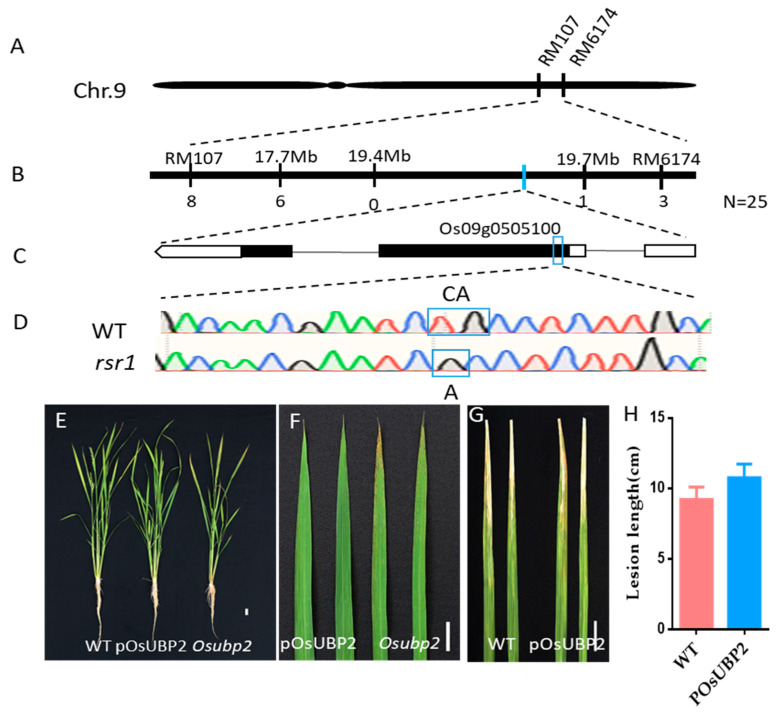
Map-based cloning of *OsRSR1*. (**A**) *OsRSR1* was initially mapped on chromosome 9 between simple sequence repeat markers RM107 and RM614. (**B**) The *OsRSR1* locus was fine-mapped to a 2-Mb genomic region between 17.7 Mb and 19.7 Mb. The numbers under the linkage map represent the number of recombinants, *n* = 25. (**C**) Structure of *OsRSR1* gene and mutation site. The line represents the intron; black boxes represent the exons; white boxes represent 3′UTR (left) and 5′UTR (right); the blue box represents the mutation site. (**D**) The sequence of the mutation site in WT and *rsr1* mutant, the *rsr1* mutant missing a base C compared with WT cased frameshift mutation. (**E**) The phenotype of 2-month-old WT, *Osubp2* mutant and pOsUBP2 T2 plants. T2: T2 generation of the pOsUBP2 transgenic plants (after selfing). (**F**) Leaves (second from the top) of pOsUBP2 T2 plants and *Osubp2* mutant at 60 days post-sowing (dps). (**G**) Leaves of WT and pOsUBP2 T2 plants were inoculated with compatible *Xoo* isolate P6. Leaves were photographed at 7 dpi. Scale Bar = 2 cm. (**H**) Lesion length on the leaves of WT and complement pOsUBP2 T2 plants at 7 dpi shown in (**G**). Error bars represent + SD (*n* = 9).

**Figure 3 plants-11-02568-f003:**
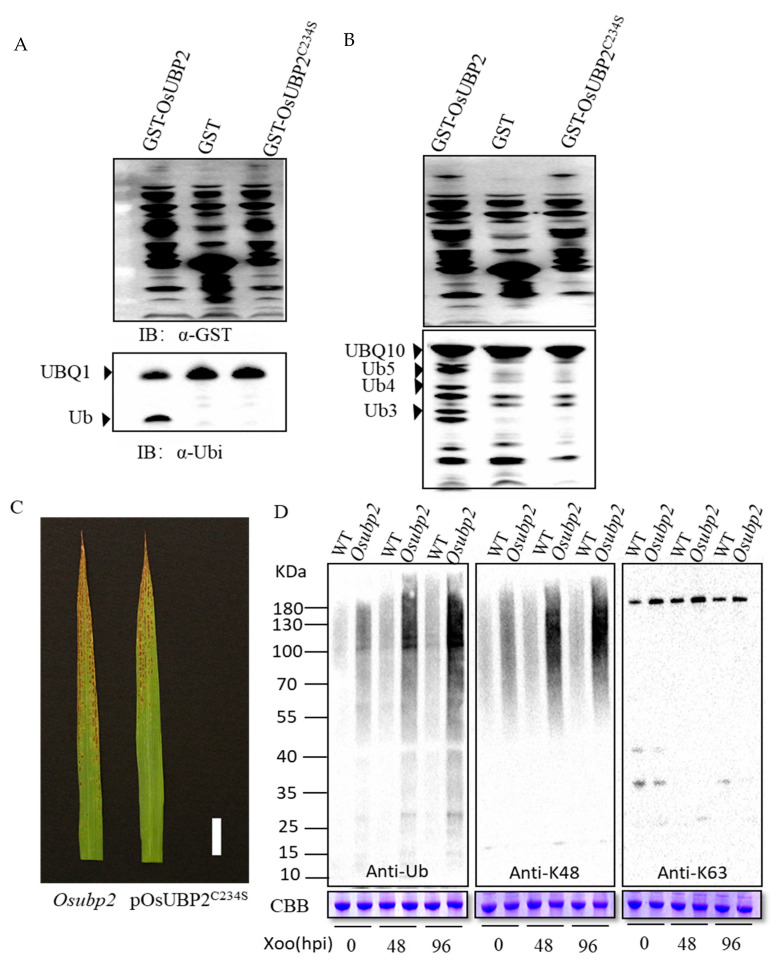
Characterization of OsUBP2 protein. (**A**,**B**) show the DUB enzyme activity of OsUBP2 and OsUBP2^C234S^ protein in vivo, detected by the degradation of the substrate AtUBQ1 in (**A**) and AtUBQ10 in (**B**) into small fragments. IB, Immunoblot; Ub, ubiquitin. (**C**) Leaves of *Osubp2* and POsUBP2^C234S^ transgenic T1 plants in *Osubp2* background at the heading stage. (**D**) The ubiquitin conjugates in WT and *Osubp2* mutant before and after *Xoo* isolate P6 inoculation. Total protein extracts from seedlings before or after *Xoo* infection were analyzed using anti-ubiquitin (Ub), anti-K48-linked (K48) or anti-K63-linked (K63) ubiquitin chain antibodies. CBB, Coomassie blue staining, hpi: hours post-inoculation.

**Figure 4 plants-11-02568-f004:**
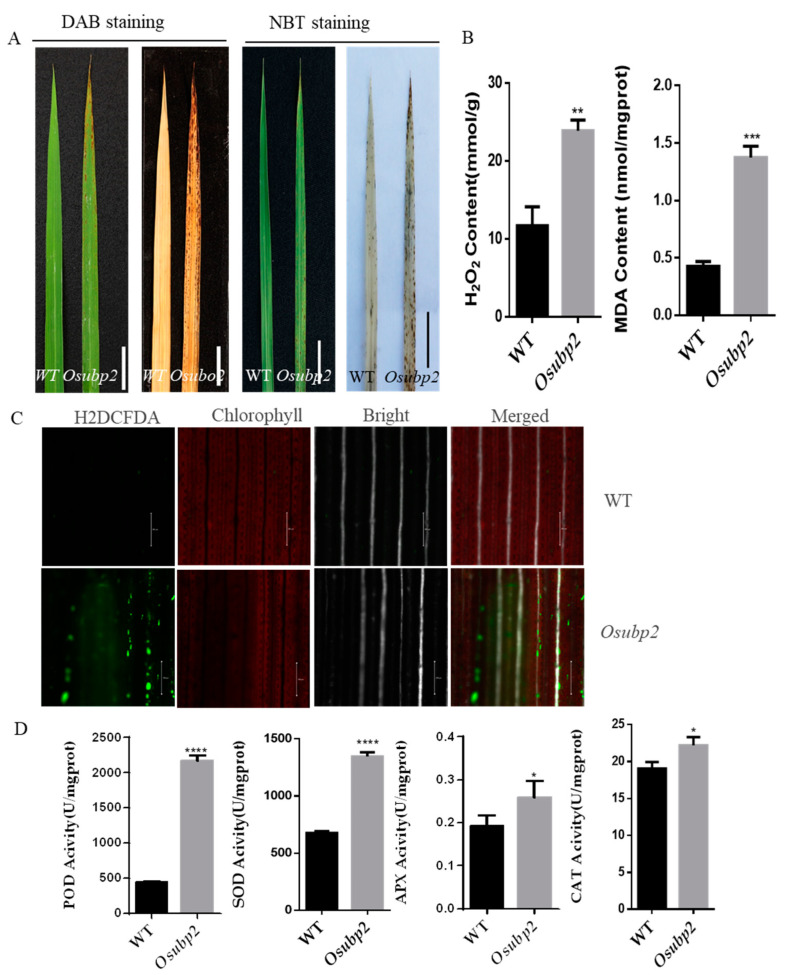
Mutation of OsUBP2 led to reactive oxygen species (ROS) accumulation. (**A**) 3,3′-diaminobenzidine (DAB) and nitro blue tetrazolium (NBT) staining of the wild type (WT) and *Osubp2* mutant leaves at 30 days post-sowing (dps). Scale Bar = 2 cm. (**B**) The content of hydrogen peroxide (H_2_O_2_) and malondialdehyde (MDA) in WT and *Osubp2* mutant leaves at 30 dps, mgprot: protein per milligram. (**C**) Microscopic analysis of leaves of the WT and *Osubp2* mutant leaves at 30 dps, incubated with 2′,7′-dichlorofluorescein diacetate (H_2_DAFDA). Red, chlorophyll; green, oxidized H_2_DCFDA. Scale Bar = 200 μm. White: white light channel; Merged: image color merged channel. (**D**) The peroxidase (POD), superoxide dismutase (SOD), ascorbate peroxidase (APX) and catalase (CAT) enzyme activity in WT and *Osubp2* mutant leaves at 30 dps. Error bars represent + SD (*n* = 3 independent pools of leaves, three plants per pool). Asterisks indicate significant difference between *Osubp2* mutant and WT by Student’s *t*-test: **** *p* < 0.0001; *** *p* < 0.001; ** *p* < 0.01; * *p* < 0.05.

**Figure 5 plants-11-02568-f005:**
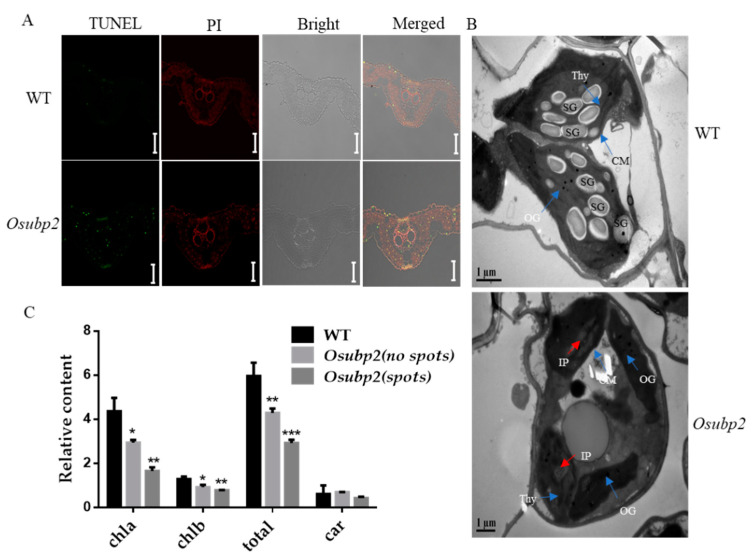
Mutation of OsUBP2 led to DNA damage and abnormal chloroplast morphology. (**A**) TUNEL assay in WT leaves and *Osubp2* mutant leaves, red signal indicates PI (Propidium iodide) staining; green represents TUNEL-positive signals. White: white light channel; Merged: image color merged channel, Scale Bar = 50 μm. (**B**) Chloroplast ultrastructure in WT (up) and *Osubp2* mutant (below). Thy: thylakoid; CM, chloroplast membrane; SG, starch granule; OG, osmiophilic globules; IP, irregular protrusions; Scale Bar = 1 μm. (**C**) Chlorophyll content in the flag leaves of WT and *Osubp2* mutant plants at 30 days post-sowing (dps). chla: chlorophyll a; chlb: chlorophyll b; car: carotene; Error bars represent + SD (*n* = 3 independent pools of leaves, three plants per pool). Asterisks indicate significant difference between the *Osubp2* and WT by Student’s *t*-test, *** *p* < 0.001; ** *p* < 0.01; * *p* < 0.05.

**Figure 6 plants-11-02568-f006:**
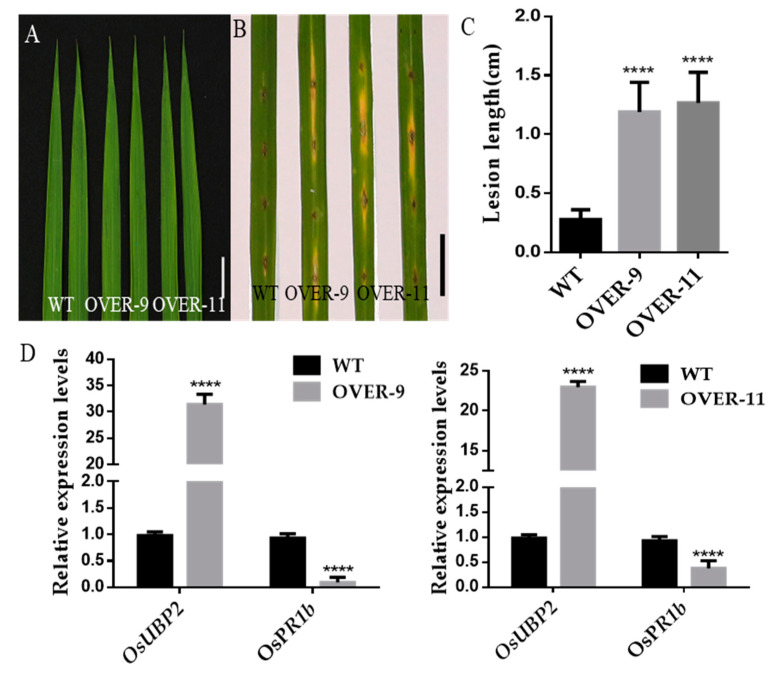
Disease resistance phenotype of *OsUBP2* overexpression plants. (**A**) Leaves of WT and *OsUBP2* overexpression plants at the heading stage. Bar = 2 cm. (**B**) Punch inoculation of *OsUBP2* overexpression plants and WT plants with the compatible *M. Oryzae* isolate RB22, Leaves were photographed at 7 dpi. Scale Bar = 2 cm. (**C**) Lesion area of the inoculated leaves of *OsUBP2* overexpression plants and WT plants shown in (**B**) at 7 dpi. Error bars represent + SD (*n* = 9). (**D**) Relative expression levels of *OsUBP2* and *OsPR1b* in two overexpression lines. Data were normalized to the expression level of the reference gene *OsACTIN*. Error bars represent +SD (*n* = 6 independent pools of leaves, two plants per pool). Asterisks indicate a significant difference between the *OsUBP2* overexpression and WT plants by Student’s *t*-test: **** *p* < 0.0001.

**Figure 7 plants-11-02568-f007:**
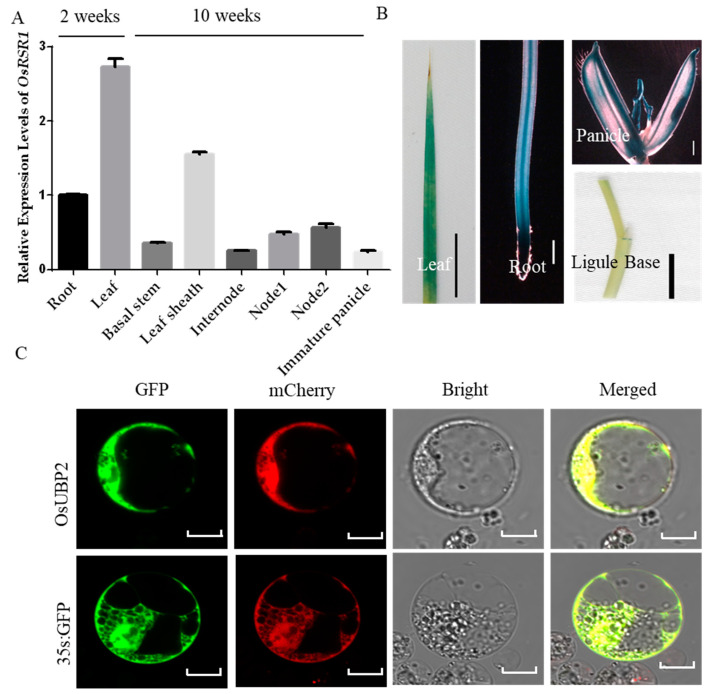
Expression pattern and subcellular location of OsUBP2. (**A**) Relative expression level of *OsUBP2* in various organs at different growth stages. Error bars represent + SD (*n* = 3). (**B**) β-glucuronidase (GUS) staining of *pOsUBP2::GUS* transgenic plants. Scale Bar = 2 cm in Leaf, Root and Ligule Base; Scale Bar = 0.2 cm in Panicle. (**C**) Subcellular localization of the OsUBP2 protein. Scale Bar = 20 μm. White: white light channel; Merged: image color merged channel.

**Figure 8 plants-11-02568-f008:**
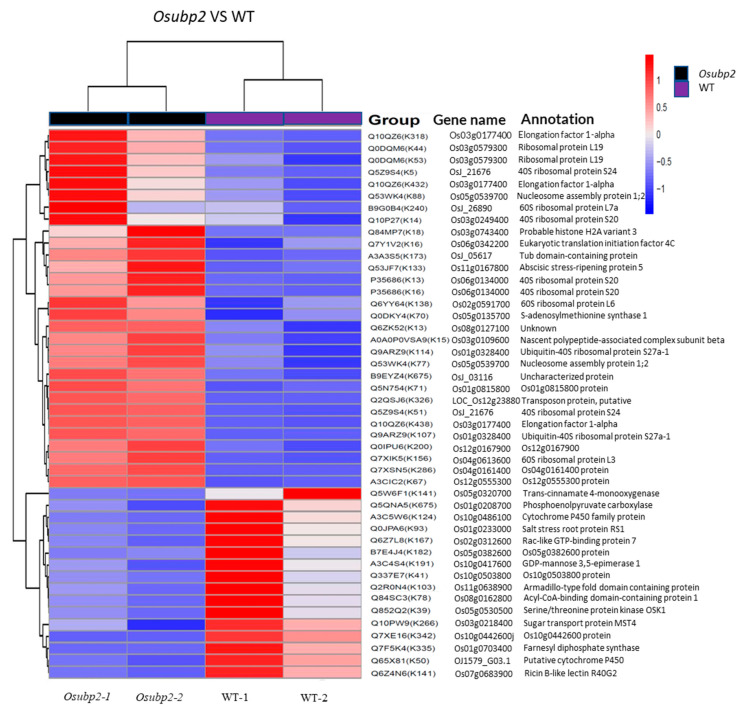
Proteomics analysis of the *Osubp2* and WT plants. Clustering result of modified sites with significant differences between the *Osubp2* and WT. WT−1 and WT-2, *Osubp2*−1, *Osubp*−2 represent two replicates. Annotation on the right side contains group (protein ID + modification site) gene name and the annotation of the gene. The information of protein ID, gene name, and the annotation of the gene was from UniProt (https://www.uniprot.org/ (accessed on 20 August 2022)). Red represents higher ubiquitination modification, and blue, lower ubiquitination modification.

## Data Availability

Not applicable.

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
