# Peer review of "Ubiquitin-Specific Protease 2 (OsUBP2) Negatively Regulates Cell Death and Disease Resistance in Rice"

_plants, 2022, doi:10.3390/plants11192568_

Round 1
Reviewer 1 Report
2022-SEP-08
Review opinion on:
Manuscript ID: plants-1899998
Title: Ubiquitin-Specific Protease 2 (OsUBP2) Negatively Regulates Cell Death and Disease Resistance in Rice
Journal: Plants
General comments:
The manuscript described the mechanisms on LMM rsr1 rice mutant. The experimental set-up is scientifically sound, advance methodology was used, and the results obtained are rich. The data presented in this manuscript should be interesting for scientists working on LMM in rice, and it adds new knowledge on mechanisms causing LMM. However, the quality of the manuscript writing is very poor. In “Material and Methods” section, many parts of the text were not represented by full sentences. “Results” section was mixed with descriptions on discussion or introductory-like. “Discussion” section was mixed with description of the results. Discussion should be focused on interpretation of results obtained in this study, not make a general reviewing. The manuscript cannot be accepted for publication in its current form.
Some specific comments are listed below.
Line 9: What “biological mechanisms” mean? A phenotype is observed on biological level, but the mechanisms correlated to a phenotype could be on molecular level or other levels. The word “biological” should be deleted, unless the authors give reasonable explanation on “biological mechanisms”.
Line 13: the pathogen type causing the disease “leaf blast” should be indicated.
Line 15 (Line 85-86): all words in the full name for “OsUBP2” must be capitalized? But in Title, was not.
Line 19: the word “in vitro”, in my impression, very often it is written “in italic”
Line 20: The authors should underline the “ubiquitinated proteomics” was done on which object? The rice LMM rsr1 or others? The word “Our” is misleading and meaningless, should be omitted.
Line 111-112, line 428-429: the rsr1 mutant was generated in the study or not? If not, the citation for rsr1 mutant should be given. If yes, should describe details of method for producing it.
Line 138: “code-shifting”? or open reading frame (ORF) shift?
Line 191: why citation [28] appeared here?
Line 201-203: this part should be in “Discussion” section.
Line 217: Many…….[2]: this part should be in “Introduction” or “Discussion” sections, should not be in “Results” section.
Line 232-233: “suggesting………”, such description should be “Discussion” section.
Line 247-250: “Plants……..[30]”, this part should not be in “Results” section.
Line 287, often in literature, the term “days post inoculation (dpi)” was used.
Line 305-306: “Next……..vector”, not accurate. The transgenic plants should contain the transgene. The gene construct in a vector was used to transform a plant. Also please check the way for writing transgene construct (to my knowledge, should be pPROMOTER::GENE in italic, “p” means promoter. The gene encoding “GUS”, includes “uidA”.
Line 310-313, “In……….benthamiana”, not clear.
Line 313-314: Fig. S5 showed results in N. benthamiana. Where are the results in “rice protoplast”? Maybe the results of “rice protoplast” were from other previous published paper (If so, “……consistent with……” should be in “Discussion” section). Again, the same problem as previously mentioned, “description of discussion” is mixed in “Results” section.
Line 325-331: should be in “Introduction” or “Methods” sections.
Line 349-350: citation(s) needed.
Line 346-373, a general reviewing on mechanisms related to LMM is placed here. The author should focus on and underline the mechanism(s) revealed by this study.
Line 363: Does the “metabolic pathway disorder” relate to the results obtained in this study?
Line 410: “Sun et al. 2022”?
Line 375-425: In “Discussion” section, many parts were repeating the results, even with indication on Figures. Such repeating should avoid.
Line 439: “A single…….gene”, why here? It seems to me should be data in “Results” section.
Line 433-439: mixed with Results, and too general.
Line 461-462: “Plasimid…………callus”, it seems not accurate, please check. To my knowledge, plasmid containing the transgene construct should be first transformed in Agrobacterium cells, then the transformed-Agrobacterium will be used to transform the explants or plant tissues. Agrobacteroum tumegaciens? Or other Agrobacterium?
Line 468: “Three……..per sample”, it is not a sentence.
Line 489: should be days post inoculation (dpi)
Line 494-495: please check grammar.
Line 506: the term “reference gene” was used more often.
Line 509: what does it mean “dps”?
Line 518-552: All the sequences mentioned in this section were sequenced in this study by the authors? If not, must clearly indicate which are done by authors in this study, which are just reference sequences taken from certain database(s).
Line 524: Twice S1?
Line 542, Line 650, repeated references.
Line 150, legend for “G” should be descript separately, it cannot be mixed with F. The current F and G, in fact indicates only F. Legend for G is missing. “and (G)” should be deleted, it is misleading, since in Line 152, the legend for F is given.
Line 153, H and I, see comments for Line 150.
Line 156: should give full description for “dps”.
Line 157-158: should use the term “reference gene”, see previous comment. Also here the authors used the term “constitutive expressed”, but in other place used the term “internal control” for the same gene, this is not allowed in one manuscript.
Line 160: RT here does not mean “real-time”. RT should mean for “reverse transcription”, gene expression level should be checked using RNA as template, so it is not PCR, but RT-PCR, in which RT means “reverse transcription”. Also, in “J”, since it is a relative quantification, should use “relative expression level (REL)”, not just “Expression level….”.
Line 180: here the authors used the term “frameshift”, but in the text “code-shift” was used. Should uniform the term used, to my knowledge, “frameshift” is accurate.
Line 181: “second on the top” is not correct. Where is WT?
Line 182: T2 was used here. Should give explanation on T2. The Fig should be understandable alone without referring to the text. T2 usually means the T2 generation of the transgenic plants (after selfing)
Line 182, in G, where is WT?
Line 183-185: give full name of “NIP”.
Line 210: “in vivo” usually was written “in italic”
Line 212-215, in D, the number 0.48 and 96, seems to me should be hours post inoculation (hpi). So, in D, better replace h as hpi, and explain hpi in legend. Line 213, “infection”. In this place should use “inoculation”. Whether the inoculated object(s) will be infected or not, is not known in the inoculation moment.
Line 236-244: give full description for “ROS, DAB, NBT and WT, dps……”. See previous comments (The Fig should be understandable alone without referring to the text). Line 238-240 in C, what is the meaning of “Bright” and “Merge”, should explain in legend. In B and D, give the meaning on “mgprot”
Line 269-277, Fig. 5, in A, explain “Bright” and ‘Merge’ in legend, see previous comments for Fig. 4, and (The Fig should be understandable alone without referring to the text). B Line 271-271, give full description for “Thy”, “OM or CM”. Line 274, give full description for dps. In C, give full name for “chla” “chlb” “car”.
Line 293-296, legend for B and C should be written independent (to avoid un-necessary misleading).
Line 296-300, should use terms “relative expression levels (REL)” and “reference gene”. See previous comments.
Line 319-322, in title of the Fig, only the first letter of the first word should be capitalized. In figure itself part A, should also use term “relative expression level(s)”. In B, Gus should be replaced by GUS, and give full name of GUS. Line 321, .please check the way on how to write transgene construct, see also previous comments. In B leaf, why the color is still green? (not paled or stained with blue color? Please check the protocol for GUS staining.) In C, explain “GFP” “mCherry” “Bright” and “Merge” in legend.
Line 341-344, Figure 8: “A” was shown on upper-left. Did Fig. 8 contain part B or C?
Meaning of the colors: in Figure Legends, Red represents higher modification, and Blue lower modification. But in the upper-right, the Red color also indicates WT; Blue-green indicates Osubp2. Does the color for WT and Osubp2 also indicates “average?” modification degrees? If not, the color for WT and Osubp2 should change into different colors than “Red and Blue-like”. The meaning for WT-1 and WT-2, Osubp2-1,-2 should be given in Legends (2 replicates ? or what). The Fig should be understandable alone to readers without referring to the text. “protein ID + modification site + gene name” is not accurate. Is “protein ID + modification site” for “Group” in the figure? But under the “Group”, there is no gene name. Gene name is another column (the source of database should be given). The authors should give the sources of the protein ID, from which database, NCBI or other? Line 342, check grammar, not clear and not accurate meaning.
Line 444: “for rice transgene”, not clear. Is it “for rice transformation”?
Line 426: In the whole section of “Materials and Methods”, in many cases, the phrases were just put together without forming a sentence, e.g., 429-431, etc…… Such quality of writing is not qualified for being reviewed. The English grammar should be intensively checked, e.g., Line 431-432, etc……
Line 389-392, “In conclusion,….”, conclusion in this place? It might be a summarizing for section 3.1? Anyway, “In conclusion…” is misleading. The conclusion is usually in the end the Discussion section, depending on the journal.
Reviewer 2 Report
The manuscript untitled " Ubiquitin-Specific Protease 2 (OsUBP2) Negatively Regulates Cell Death and Disease Resistance in Rice" is presenting a great amount of work concerning the deregulation of expression of OsUBP2 showing a rolein immune processes and ROS production.
Despite a very well done scientific approched, I have a huge concern concerning the originality of this work in comparison to "Sun, J.; Song, W.; Chang, Y.; Wang, Y.; Lu, T.; Zhang, Z. OsLMP1, encoding a deubiquitinase, regulates the immune response in rice. Front. Plant Sci. 2022, 12."
First, this later reference should be presented in introduction and not in result or discussion.
As far I might understand OsLMP1=OsUBP2, then this work present numerous similar data and figure with the work of Sun and al.
Figure 1 is similar with same phenotype, Figure 3 is similar to Fig 6A-C, Figure 6 is similar to figure 4.
Minor points:
Magnaporthe oryzae and Xanthomonas oryzae pv. oryzae = Xoo should be presented elsewere than in Abstract, at least firstly in lines 121-122
Lines 140-142 are confuse to me as OsLMP1 was already founded close to AtUBP2
Legend of Fig 1:
Change Express Level of XXX to Relative expression
Line 151: Change to M. oryzae
Legend of Fig 2:
In H, replace NIP by WT, replace POsUBP2 bu pOsUBP2
Lines 201-203, should be inserted in the introduction
Figure 3: What is DUB activity, and replace POsUBP2 bu pOsUBP2
Figure 4: B, Change axe to MDA content....
As specified in line 252, a nuclei localisation might need a presentation of a magnfied plant cell.
Figure 5: What is Thy ?
Figure 6: Change to M. oryzae in line 294
Figure 7 is similar to previous work.
Line 338, missing figure 8 analysis
Figure 8, I do not understand annotation, for example Elongation factor 1-alpha appears 3 times with diverse regulation profile.
Line 409-411 move to introduction
Fig S3, Change NIP to WT
Fig S4, what are representing number in the alignement
Reviewer 3 Report
Comments: The manuscript entitled "Ubiquitin-Specific Protease 2 (OsUBP2) Negatively Regulates Cell Death and Disease Resistance in Rice" by Jiang et al. identified a rice Lesion mimic mutants (rsr1) that showed an enhanced resistance to both leaf blast, and bacterial blight, etc. Also, OsUBP2 has been found as a negative regulator of immune processes and ROS production. The introduction gives useful background on the approach. The results were well studied, and the discussion was reasonable. In order to improve the paper, I have the following comments on the current version of the manuscript.
Main issues:
In the cloning of OsRSR1, the QTN mapping results or other evidence need to add as important evidence for identification of OsRSR1.
Minor concerns,
1. Line 55, The introduction of “Lesion mimic mutants” should be moved to the head of the “Introduction”.
2. Line 166, (Figure 2G, H). is (Figures 2G, 2H).
3. Line 284, “…than in...” is “…when compared with...”
4. Line 310, “…OsUBP2...” is “…OsUBP2...”
5. Line 334, “a P-value” is P-value
6. Take care of langue-related issues.
Round 2
Reviewer 2 Report
The manuscript " Ubiquitin-Specific Protease 2 (OsUBP2) Negatively Regulates Cell Death and Disease Resistance in Rice Authors: Xiaorong Mo *, Ruirui Jiang, Shichen Zhou, Xiaowen Da, Tao Chen, Jiming Xu, Peng Yan" is now really good for publication after the changes in the previous manuscript.
All points have been adressed by authors.
Il's a very interesting paper that will completed the work of team (Sun, J.; Song, W.; Chang, Y.; Wang, Y.; Lu, T.; Zhang, Z.)
Minor points :
Need reference [27] at line 118
There is still some typo in the review manuscript
